# Risk Factors for Nodding Syndrome and Other Forms of Epilepsy in Northern Uganda: A Case-Control Study

**DOI:** 10.3390/pathogens10111451

**Published:** 2021-11-09

**Authors:** Nolbert Gumisiriza, Marina Kugler, Nele Brusselaers, Frank Mubiru, Ronald Anguzu, Albert Ningwa, Rodney Ogwang, Pamela Akun, Amos Deogratius Mwaka, Catherine Abbo, Rogers Sekibira, An Hotterbeekx, Robert Colebunders, Kevin Marsh, Richard Idro

**Affiliations:** 1Department of Mental Health, Kabale University School of Medicine, Kabale P.O. Box 317, Uganda; gumag5@gmail.com; 2Global Health Institute, University of Antwerp, 2600 Antwerp, Belgium; marina.mbc@gmx.de (M.K.); Nele.Brusselaers@uantwerpen.be (N.B.); an.hotterbeekx@uantwerpen.be (A.H.); robert.colebunders@uantwerpen.be (R.C.); 3Centre for Translational Microbiome Research, Karolinska Institute, 17177 Stockholm, Sweden; 4Department of Statistical methods, School of Statistics and Planning, College of Business and Management Sciences, Makerere University, Kampala P.O. Box 7062, Uganda; mubiruf09@gmail.com; 5College of Health Sciences, Makerere University, Kampala P.O. Box 7072, Uganda; ranguzu@mcw.edu (R.A.); albert.ningwa@gmail.com (A.N.); rjogwang@gmail.com (R.O.); pamakun@gmail.com (P.A.); Amos.mwaka@mak.ac.ug (A.D.M.); cathya180@gmail.com (C.A.); sekibira@gmail.com (R.S.); 6Division of Epidemiology, Institute for Health and Equity, Medical College of Wisconsin, Milwaukee, WI 53226, USA; 7Centre of Tropical Neuroscience, Kitgum Site, Kampala P.O. Box 27520, Uganda; 8KEMRI-Wellcome Trust Research Programme, Centre for Geographic Medicine Coast, Kilifi P.O. Box 230-80108, Kenya; 9Centre for Tropical Medicine and Global Health, University of Oxford, Oxford OX3 7LG, UK; kevin.marsh@ndm.ox.ac.uk

**Keywords:** nodding syndrome, epilepsy, onchocerciasis, Uganda, risk factors

## Abstract

Epidemiological studies suggest a link between onchocerciasis and various forms of epilepsy, including nodding syndrome (NS). The aetiopathology of onchocerciasis associated epilepsy remains unknown. This case-control study investigated potential risk factors that may lead to NS and other forms of non-nodding epilepsy (OFE) in northern Uganda. We consecutively recruited 154 persons with NS (aged between 8 and 20 years), and age-frequency matched them with 154 with OFE and 154 healthy community controls. Participants’ socio-demography, medical, family, and migration histories were recorded. We tested participants for *O. volvulus* serum antibodies. The 154 controls were used for both OFE and NS separately to determine associations. We recruited 462 people with a median age of 15 years (IQR 14, 17); 260 (56.4%) were males. Independent risk factors associated with the development of NS were the presence of *O. volvulus* antibodies [aOR 8.79, 95% CI (4.15–18.65), *p*-value < 0.001] and preterm birth [aOR 2.54, 95% CI (1.02–6.33), *p*-value = 0.046]. Risk factors for developing OFE were the presence of O. volvulus antibodies [aOR 8.83, 95% CI (4.48–17.86), *p*-value < 0.001] and being born in the period before migration to IDP camps [aOR 4.28, 95% CI (1.20–15.15), *p*-value = 0.024]. In conclusion, *O. volvulus seropositivity* was a risk factor to develop NS and OFE; premature birth was a potential co-factor. Living in IDP camps was not a risk factor for developing NS or OFE.

## 1. Introduction

Nodding syndrome (NS) is a neurological disorder characterized by repeated head-nodding. The symptoms develop in early childhood, and patients progressively develop neurocognitive and physical impairment [1]. It has been suggested that NS is one of the clinical presentations of onchocerciasis-associated epilepsy (OAE) [2,3,4].

Between 1986 and 2006, northern Uganda experienced a civil war that disrupted the region’s security, socioeconomic activities, and health programs. For safety, people migrated into vast, densely populated internally displaced people (IDP) camps. The sanitation, housing, and general health conditions in IDP camps were poor, leading to several disease outbreaks [5]. The period of civil war and IDP camps coincided with the peak incidence of NS and other forms of epilepsy. This led to several hypotheses that directly or indirectly linked the etiopathology of NS to the war and IDP camp conditions [6,7,8,9,10]. Besides the epidemiological linkage between *O. volvulus* and NS, none of the hypotheses about the association between NS and IDP camp conditions has been confirmed so far [11,12].

The NS epidemic in Uganda started between 1998 and 2000, mainly in three onchocerciasis-endemic districts: Kitgum, Pader, and Lamwo [13]. This epidemic stopped in 2014/2015, after implementing the bi-annual community-directed treatment with ivermectin (CDTI) and larviciding the major rivers [14]. A decrease in other forms of non-nodding epilepsy (OFE) was also noted in the same period [15]. In 1994, an NS-like disorder was reported in the onchocerciasis-endemic region in western Uganda, but it disappeared following the elimination of onchocerciasis [16]. Several epidemiological studies concerning epilepsy in other onchocerciasis-endemic regions strongly suggest that infection with *O. volvulus* is linked to the development of epilepsy [17,18,19,20,21]. However, the pathological mechanisms by which infection with *O. volvulus* triggers epilepsy remains unknown [22,23]. Neither *O. volvulus* microfilariae DNA nor *Wolbachia* DNA has been detected in the cerebrospinal fluid of persons with NS [23,24]. Furthermore, a post-mortem study did not show evidence of the parasite nor *O. volvulus* microfilariae DNA in the brain [25].

NS only occurs in onchocerciasis-endemic regions. However, not all *O. volvulus* infected persons develop NS, and only children between the ages of 3–18 years develop NS. The level of *O. volvulus* infection seems to be a risk factor for developing epilepsy in onchocerciasis-endemic areas [26]. However, even with a similar *O. volvulus* microfilarial density, some children develop epilepsy, including NS, and others not. Therefore, other co-factors such as nutritional, co-infections [12] and genetics [27] may play a role. Here, we examined potential risk factors, including demographic, socioeconomic, onchocerciasis related, medical history, family characteristics, and life in IDP camps potentially associated with the NS and OFE in northern Uganda.

## 2. Materials and Methods

### 2.1. Design

This was a case-control study nested within a clinical trial in northern Uganda (NCT02850913). The clinical trial aimed to determine the therapeutic effect of doxycycline in children with NS and investigate whether NS was a neuro-inflammatory disorder induced by *O. volvulus* or its co-symbiotic bacteria, *Wolbachia* [28,29].

### 2.2. Setting

The study was conducted in the NS endemic districts of Kitgum, and Pader, in northern Uganda. The region is vastly rural, practicing subsistence agriculture. The area is undergoing socioeconomic rehabilitation following a protracted civil war that had displaced many inhabitants into IDP camps [30]. Specialized treatment centers and community outreach clinics were set up following the NS epidemic across the affected districts. These NS centres also cared for patients with OFE [31]. Individuals recruited into the study were transported to Kitgum general hospital for comprehensive assessments [28]. The hospital is located 450Km from the capital city and cared for 1321 of the 3320 documented children with NS in Uganda [31].

### 2.3. Participants

Between September 2016 and September 2017, the first 154 consecutive persons with NS to be enrolled in the doxycycline clinical trial were recruited into the case-control study. Two sets of age-frequency matched controls, 154 with OFE and 154 healthy community controls, were recruited as a comparison. A case of NS was defined using the WHO criteria: head nodding on two or more occasions, occurring in clusters at a frequency of 5–20/minute; onset of nodding seizures between 3 and 18 years; observed by a trained health worker or documented on EEG/EMG; plus any one of: (a) triggered by food or cold weather; (b) presence of other seizures or neurological abnormalities and cognitive decline; and (c) clustering in space or time [32]. Other forms of epilepsy (OFE) included all varieties of epilepsy that met the International League Against Epilepsy criteria (two or more unprovoked seizures occurring at least 24 hours apart) [33], but that did not involve nodding of the head. The healthy controls were persons in the same community as the NS and OFE cases, who have never suffered from NS nor OFE.

### 2.4. Data Collection

In collaboration with their attendants, the study participants responded to a standardized case report form containing questions relating to possible risk factors of NS and OFE. Questions were asked about socioeconomic and poverty indices (education, housing conditions, possessions, source of income), health facility and behavior, IDP camp settlements and conditions, complications during the antenatal, delivery, immediate postnatal period, and early infancy. Exposure to *O. volvulus* was determined for each participant using the anti–OV16 IgG levels in plasma. These tests were conducted at the USA National Institute of Health’s laboratory of parasite research in Bethesda. A positive test was defined as an anti–OV16 IgG signal-to-noise ratio greater than 2 [34]. From each participant, two skin-snips were obtained from the iliac crests to be examined for the presence of *O. volvulus* microfilariae. The data collected was entered in the Epi info database then exported to STATA version 15.0 for further management and analysis.

### 2.5. Statistical Analysis

We described participant characteristics using counts and percentages. We compared the differences in these characteristics between cases and controls using the Pearson and Fisher’s exact chi-square tests, and the Kruskal-Wallis rank test. The 154 controls were used for both OFE and NS separately to determine associations. We fitted an ordinary logistic regression to assess risk factors for developing NS and risk factors for having OFE. Factors with a *p*-value ≤ 0.25 at bi-variable analysis were entered into a multivariable ordinary logistic regression model. Variables with a *p*-value ≤ 0.05 on multivariable logistic regression were considered statistically significant risk factors for developing NS and OFE.

## 3. Results

### 3.1. Socio-Demographic Characteristics of the Study Population

In total, 462 participants were recruited in the study: 154 persons with NS, 154 with OFE, and 154 healthy controls. One community control was excluded from this analysis due to an incomplete assessment. Males were 56.4% (260/461) of the study population (Table 1). There were 227 participants (49.2%) from Kitgum and 234 (50.8%) from Pader. The median age was 15 years (IQR; 14–17), with 80.7% being aged between 14 and 17 years. The median number of siblings per family was five (IQR; 4-7). The median monthly household income was UGX 20,000 (IQR; 10,000, 50,000). At some time in their life, 93.3% of all research participants resided in IDP camps. The median duration spent in IDP camps for persons with NS and OFE was five years, and six years for controls (Table 2). One-hundred thirteen of one hundred fifty-four persons with NS (73.4%, *p*–value < 0.001) and 107/154 persons with OFE (69.5%, *p*–value < 0.001) participants were born before their parents fled into IDP camps, compared to 71/153 controls (46.4%). Noteworthy, 95% of the study participants had received a dose of ivermectin.

### 3.2. Risk Factors Associated with Nodding Syndrome

Preterm births were reported more frequently among persons with NS [24/154 (15.6%) *p*-value = 0.003] compared to healthy controls [8/153 (5.2%)] (Table 2). The preterm births were significantly associated with NS [aOR 2.54, 95% CI (1.02–6.33), *p*-value = 0.046] (Table 3).

More than 90% of NS cases [144/154 (93.5%)] were seropositive for *O. volvulus*, compared to 54.9% (84/153) of the controls. The median anti–OV16 IgG signal-to-noise ratio in NS cases was 26 (IQR; 8, 43) compared to 2 (IQR; 1, 10) in community controls (*p*-value = 0.001) (Table 2). The *O. volvulus* seropositivity was also significantly associated with NS [aOR 8.79, 95% CI (4.15–18.65), *p*-value < 0.001] (Table 3). *O. volvulus* seropositivity also increased with increasing age: in the age group 6–13 years, 39 (60.9%) of the 64 were seropositive; in the age-group 14–15 years, 144 (76.2%) of the 189; in the age-group 16–17 years, 163 (89.1%) of the 183; and in the age group 18–24 years 23 (92.0%) of the 25. *Onchocerca volvulus* microfilariae were detected in skin snips in 8.4% of persons with OFE, 5.2% of persons with NS and 1.3% in healthy controls (Table 2).

A history of severe malaria was less frequent among persons with NS [26/154 (16.9%)] than in controls [43/153 (28.1%), *p*–value = 0.019]. Only 12.3% (19/154) of the NS cases reported a history of severe malaria in the years before the onset of the nodding seizures. Blind family members of persons with NS were younger [median 40 years (IQR 18,48)] than blind family members of controls [median age 50 years (IQR 40,70)], *p*–value = 0.004.

### 3.3. Risk Factors Associated with Other Forms of Epilepsy

Like for NS, preterm births were reported more frequently among children with OFE [20/154 (13.0%) *p*–value = 0.018] compared to healthy controls [8/153 (5.2%)] (Table 2), however this was not significantly associated with OFE [aOR 1.77, 95% CI (0.69–4.58), *p*–value = 0.236].

More persons with OFE [141/154 (91.6%)] were seropositive for *O. volvulus* compared to healthy controls [84/153 (54.9%), *p*–value < 0.001]. The median anti–OV16 IgG signal-to-noise ratio among persons with OFE was 18 (IQR, 6, 30) compared to 2 (IQR, 1, 10) in controls (*p*–value = 0.001) (Table 2). There was a significant association between OFE and O. *volvulus* seropositivity [aOR 8.83, 95% CI (4.48–17.86), p-value < 0.001]. Furthermore, we observed that being born before the parents relocated to IDP camps was a risk factor for developing OFE [aOR 4.28, 95% CI (1.20–15.15), *p*-value = 0.024] (Table 3). Among the other factors measured, only the birth period and history of ivermectin use were significantly different between individuals with OFE and healthy controls (Table 2). There was, however, no substantial association with OFE (Table 3). 

## 4. Discussion

We conducted a case-control study to identify the risk factors for the development of NS and OFE in northern Uganda. We observed that *O. volvulus* seropositivity was associated with both NS and OFE. The study also showed preterm birth as a possible risk factor of acquiring NS. Contrary to previous reports, we observed that being born in IDP camps was not a risk factor for either NS or OFE.

Our findings are in line with previous case-control studies, which also showed that persons with NS and OFE in onchocerciasis-endemic regions were more likely to present *O. volvulus* antibodies than controls [19,20,21,35]. Relatedly, blind family members of NS participants were generally younger than those of controls. This finding suggests a higher past exposure to *O. volvulus* infected blackflies in families with persons having NS. Although there was high seropositivity for *O. volvulus* among both NS and OFE cases, the median anti–OV16 IgG signal-to-noise ratio was higher among persons with NS [26 (IQR; 8, 43)] compared to those with OFE [18 (IQ; 6, 30)]. The low numbers of study participants with positive skin snips (presence of active onchocerciasis infections) can be attributed to the rigorous onchocerciasis elimination programs (vector control and biannual mass distribution of ivermectin) in the study area. Our study suggests that in onchocerciasis-endemic areas, both NS and OFE might have a common etiology, but NS appears in children most exposed to *O. volvulus* infected blackflies. This corroborates with a study in an onchocerciasis-endemic area in South Sudan that showed that persons with NS present a more severe form of epilepsy with a higher microfilarial load than persons with OFE [36] and a study in an onchocerciasis endemic regions in the Democratic Republic of Congo that showed a higher frequency of seizures was associated with higher microfilarial load levels [37].

We observed an increased frequency of preterm births among persons with NS. The study participants’ guardians gave the birth history in most cases, as the majority of the participants were minors. The question asked was whether the participant was born at term (pregnancy had completed nine months). A study in Ecuador suggested that onchocerciasis may be associated with an increased risk of spontaneous abortions [38]. In this study, the effects of onchocerciasis on preterm birth were not investigated, but as spontaneous abortions have been associated with preterm birth, it is possible that onchocerciasis could lead to preterm birth [39]. Therefore, it can be hypothesized that preterm birth in northern Uganda may have resulted from an *O. volvulus* infection in the mother during pregnancy. In-utero exposure to *O. volvulus* alters the child’s immune response towards the parasite, leading to a higher microfilarial load in infected children [40]. Such a high microfilarial load, at a very young age, may have been the trigger for NS. Preterm birth may also lead to malnutrition, increasing the risk for infectious diseases and potentially a higher microfilarial load.

Previous research had suggested that environmental factors, particularly those related to IDP camps, played a role in the development of NS [8,10,19]. These factors ranged from munition, contaminated food, and water. Other studies have since debunked this theory, including this study [7,21]. However, compared to the controls, most persons with NS and OFE were born before their families migrated to the IDP camps. This period before the creation of IDP camps coincided with a deficient onchocerciasis control program in northern Uganda. This resulted in a high *O. volvulus* exposure of the children who later developed NS and OFE.

Preterm birth has been linked to epilepsy globally, and the association becomes stronger as gestational age decreases [41]. Specific to the tropics, preterm births have been related to deficient antenatal care services and prevalent maternal infections [42]. However, preterm birth as a risk factor for NS had not been reported so far. Whether this is related to an *O. volvulus* infection of the mother during pregnancy requires further investigation. If preterm birth related to an *O. volvulus* infection during pregnancy is a risk factor for NS and OFE, it may be advantageous to treat *O. volvulus* infected pregnant women with ivermectin. However, currently, pregnancy is a contra-indication for ivermectin treatment [43]. Therefore, additional studies are needed to investigate the safety and potentially beneficial effect of ivermectin in *O. volvulus* infected pregnant women.

Our study results do not support the hypothesis that the cause of NS is linked to the life in IDP camps, a belief that is still strongly present in the affected districts in northern Uganda.

This study had some limitations. A proportion of controls were siblings of NS participants, and we did not record when the control was a sibling of an NS case. We were, therefore, unable to compare NS cases with controls from other families. Secondly, the onset of seizures in all participants was many years before enrolment in the case-control study. Therefore, there might have been a recall bias, especially relating to conditions in infancy. Since this study was initiated before the definition of OAE was published, we did not systematically assess the study participants for all OAE criteria. The assessment of onchocerciasis nodules among the study population was not consistently carried out to be considered for analysis. Finally, we cannot exclude that in some participants the epilepsy was caused by neuro-cysticercosis. However, previous brain imaging and post-mortem studies did not show evidence for neurocysticercosis as a possible cause of NS and OAE in northern Uganda [24,25,44]. Moreover, in a previous case-control study in the study area, all NS cases and all controls did not present *Taenia solium* antibodies [19].

## 5. Conclusions

Our findings support the relationship between *O. volvulus* seropositivity and the development of NS and OFE in the northern Uganda study area. The study highlights the significance of boosting onchocerciasis elimination initiatives in onchocerciasis-endemic areas, where epilepsy is highly prevalent.

## Figures and Tables

**Table 1 pathogens-10-01451-t001:** Socio-demographic characteristics of the study participants.

Characteristics	Nodding Syndrome (N = 154)	Other Forms of Epilepsy (N = 154)	Healthy Controls (N = 153)	Total (N = 461)	*p*-Value
Sex, n (%)					0.466
Female	71 (46.1)	61 (39.6)	69 (45.1)	201(43.6)	
Male	83 (53.9)	93 (60.4)	84 (54.9)	260 (56.4)	
Median age in years, (IQR)	16 (14, 17)	16 (14, 17)	15 (14, 16)	15 (14, 17)	<0.001
Age distribution					0.782
6-9	3 (1.9)	2 (1.3)	3 (2.0)	8 (1.7)	
10-14	37 (24.0)	37 (24.0)	48 (31.4)	122 (26.5)	
15-18	111 (72.1)	112 (72.7)	99 (64.7)	322 (69.8)	
>18	3 (1.9)	3 (1.9)	3 (2.0)	9 (2.0)	
District, n (%)					0.490
Kitgum	79 (51.3)	79 (51.3)	69 (45.8)	227 (49.2)	
Pader	75 (48.7)	75 (48.7)	84 (54.2)	234 (50.8)	
Median monthly household income, UGX (IQR)	20,000 (10,000,50,000)	17,500 (10,000, 50,000)	20,000 (10,000, 50,000)	20,000 (10,000, 50,000)	0.287
Median number of siblings (IQR)	5 (4, 7)	6 (4, 7)	5 (4, 7)	5 (4, 7)	0.223

All comparisons made using chi-square.

**Table 2 pathogens-10-01451-t002:** Comparison of individual and environmental characteristics among healthy controls versus cases of nodding syndrome and other forms of epilepsy in northern Uganda.

Characteristics	Nodding Syndrome (N = 154)	*p-*Value	Other Forms of Epilepsy (N = 154)	*p-*Value	Healthy Controls (N = 153)
Mother experienced illness during pregnancy of participant, n (%)		0.465		0.085	
No	138 (89.6)		143(92.9)		133 (86.9)
Yes	16 (10.4)		11 (7.1)		20 (13.1)
Preterm birth ^β^, n (%)		0.003		0.018	
No	130 (84.4)		134 (87.0)		145 (94.8)
Yes	24 (15.6)		20 (13.0)		8 (5.2)
Born by normal vaginal delivery ^δ^, n (%)		0.320		0.647	
No	6 (3.9)		2 (1.3)		3 (2.0)
Yes	148 (96.1)		151 (98.7)		150 (98.0)
Period the participant was born, n (%)		<0.001		<0.001	
Before family went to IDP camps	113(73.4)		107 (69.5)		71 (46.4)
In the IDP camps	32 (20.8)		43 (27.9)		68 (44.4)
Never lived in IDP camps	9 (5.8)		4 (2.6)		14 (9.2)
Lived in IDP camps		0.659		0.094	
No	11 (7.1)		6 (3.9)		13 (8.5)
Yes	143(92.9)		148 (96.1)		140 (91.5)
Duration in IDP camps, median (IQR)	5 (3, 7)	0.042	5 (4, 7)	0.319	6(4, 8)
Anti–OV16 IgG ^€^ signal-to-noise ratio, median (IQR)	26 (8, 43)	<0.001	18 (6, 30)	<0.001	2 (1, 10)
Seropositive for *O volvulus* ^¥^, n (%)		<0.001		<0.001	
Negative	10(6.5)		13 (8.4)		69 (45.1)
Positive	144(93.5)		141 (91.6)		84 (54.9)
Positive skin snip for *O. volvulus*, n (%)	8 (5.19)	0.055	13 (8.44)	*0.004*	2 (1.31)
History of taking ivermectin, n (%)		0.548		*0.007*	
No	5 (3.2)		0(0.0)		7 (4.6)
Yes	149 (96.7)		154 (100)		146 (95.4)
Blind family member, n (%)		0.164		0.675	
No	118 (76.6)		125 (81.2)		127 (83.0)
Yes	36 (23.4)		29 (18.8)		26 (17.0)
Median age of blind family member, years (IQR)	40 (18, 48)	0.004	48 (30, 60)	0.323	50 (40, 70)
History of severe malaria, n (%)		0.019		0.293	
No	128 (83.1)		116 (75.3)		110 (71.9)
Yes	26 (16.9)		38 (24.7)		43 (28.1)

NS = Nodding syndrome; OFE = Other forms of epilepsy; IDP = Internally Displaced Persons; ^β^ Birth was less than nine (9) gestational months; ^δ^ Birth by spontaneous vaginal delivery; ^€^ Immunoglobulin G (antibody); ^¥^ A sample was considered seropositive for *O. volvulus* when anti–OV-16 IgG signal-to-noise ratio was above 2. Cases and controls were compared using the Pearson and Fisher’s exact chi-square test, and the Kruskal-Wallis rank test.

**Table 3 pathogens-10-01451-t003:** Logistic regression model of risk factors associated with having nodding syndrome, and risk factors of other forms of epilepsy in northern Uganda.

	Nodding Syndrome				Other Forms of Epilepsy			
Characteristics	Unadjusted OR (95% CI)	*p-*Value	Adjusted OR (95% CI)	*p*-Value	Unadjusted OR (95% CI)	*p*-Value	Adjusted OR (95% CI)	*p-*Value
Sex								
Female	1 (base)				1 (base)			
Male	0.96 (0.61–1.50)	0.86	1.04 (0.62–1.76)	0.868	1.25 (0.80–1.97)	0.331	1.37 (0.77–2.30)	0.302
Age distribution								
6–9	1 (base)				1 (base)		-	
10–14	0.77 (0.15–4.04)	0.758	0.54 (0.07–4.18)	0.551	1.03 (0.51–2.06)	0.939	0.56 (0.06–5.50)	0.622
15–18	1.12 (0.22–5.68)	0.890	0.49 (0.07–3.68)	0.491	3.13 (1.50–6.52)	*0.002*	0.53 (0.05–5.08)	0.578
>18	1 (0.10–9.61)	1.000	0.34 (0.02–4.76)	0.425	3.40 (1.15–10.05)	*0.027*	0.31 (0.02–5.66)	0.431
Period the participant was born								
Before family went to IDP camps	2.48 (1.02–6.02)	0.046	1.50 (0.53–4.24)	0.448	5.27 (1.67–16.67)	0.005	4.28 (1.20–15.15)	0.024 **
In the IDP camps	0.73 (0.29–1.87)	0.514	0.65 (0.22–1.94)	0.439	2.21 (0.68–7.17)	0.185	2.30 (0.63–8.40)	0.207
Never lived in IDP camps	1 (base)				1 (base)			
Preterm birth ^β^								
Yes	3.35 (1.45–7.71)	0.005	2.54 (1.02–6.33)	0.046 **	2.71 (1.15–6.35)	0.022	1.77 (0.69–4.58)	0.236
History of severe malaria								
Yes	0.52(0.30–0.90)	0.020	0.57 (0.30–1.07)	0.082	0.78 (0.47–1.30)	0.344	-	-
Seropositive for *O. volvulus* ^¥^								
Yes	11.8 (5.80–24.20)	<0.001	8.79 (4.15–18.65)	<0.001 **	8.91(4.65–17.09)	<0.001	8.83 (4.48–17.86)	<0.001 **
Blind family member								
Yes	1.50 (0.85–2.62)	0.165	1.39 (0.73–2.68)	0.319	1.13 (0.63–2.03)	0.675	-	-

Variables with *p*-value ≤ 0.25 at bi-variable analysis (unadjusted OR) proceeded to the multivariable model (adjusted analysis). ** Statistically significant on multivariable logistic regression *(p*-value ≤ 0.05); ^β^ Birth was less than nine (9) gestational months; ^¥^ A sample was considered seropositive for *O. volvulus* when anti–OV-16 IgG signal-to-noise ratio was above a cut-off of 2.

## Data Availability

The corresponding author may provide the datasets generated during the current investigation on reasonable request.

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
