# Peer review of "Risk Factors for Nodding Syndrome and Other Forms of Epilepsy in Northern Uganda: A Case-Control Study"

_pathogens, 2021, doi:10.3390/pathogens10111451_

Round 1

Reviewer 1 Report

Gumisiriza et al. performed a case control study to decipher risk factors of nodding syndrome and other forms of epilepsy (OFE) in a study site in Northern Uganda. They revealed that nodding and OFE is associated with Ov16 antibodies, whereas OFE is associated with birth before settlement in camps for internally displaced people (IDP) and nodding with preterm birth. Although the results are nicely presented and discussed some issues need to be clarified.

1) Recently the group of Colebunders published several papers about onchocerciasis associated epilepsy (OAE) and I am wondering why the criteria OFE instead of OEA was here applied.

2) Can the authors exclude neurocysticercosis in the nodding and OFE cohorts?

3) Why did the authors not compare nodding, OFE and healthy controls together? This would be a more appropriate approach especially since the control group came from the same study site. Moreover, multivariate regression analysis comparing nodding vs OFE, nodding vs healthy controls and OFE vs healthy control would be important to perform to analyse the differences between nodding and OFE. Finally, in my opinion it is redundant and not necessary to show the values of the healthy controls twice in the very same table (Table 2).

4) Can the authors explain why the age groups where separated in four different groups? What is the rational behind this separation? For example, why did the authors separate the age groups 14-15 and 16-17?

5) In regards to Ov16 antibody results, can the authors show the Ov16 IgG  single to noise ratio and seropositive numbers in the different age groups? Is there an increase of Ov16 positivity with increasing age (due to increased duration of exposure to O. volvulus)?

6) Besides the Ov16 results did the authors have data sets about Ov microfilaria counts, presence of nodules or data sets about clinical examination to elucidate active Ov infection. Presence of Ov16 antibodies do not predict if the patient was infected/have an ongoing infection since exposure to Ov without infection manifestation is also possible. If not this needs to be stated as limitation.

Author Response

We thank the Reviewer for taking the time to review, comment, and give suggestions to this paper. Your comments have been valuable and enriching to the paper. Below are the responses to your comments and suggestions.

REVIEWER 1

Gumisiriza et al. performed a case control study to decipher risk factors of nodding syndrome and other forms of epilepsy (OFE) in a study site in Northern Uganda. They revealed that nodding and OFE is associated with Ov16 antibodies, whereas OFE is associated with birth before settlement in camps for internally displaced people (IDP) and nodding with preterm birth. Although the results are nicely presented and discussed some issues need to be clarified.

1. Recently the group of Colebunders published several papers about onchocerciasis associated epilepsy (OAE) and I am wondering why the criteria OFE instead of OEA was here applied.

Response:

The term “Onchocerciasis Associated Epilepsy (OAE)” refers to the different forms of epilepsy (including Nodding syndrome and Nakalanga syndrome) that are epidemiologically linked to an Onchocerca volvulus infection and that meet specific criteria (Colebunders et al., 2018). Since this criterion was not systematically assessed in this case-control study, we did not use the OAE definition in this paper.

The term “Other forms of Epilepsy (OFE)” refers to all forms of epilepsy that do not involve nodding of the head.

In the description of participants in the study methodology, we now state:  “Other forms of Epilepsy (OFE) included all varieties of epilepsy that met the International League against Epilepsy criteria (two or more unprovoked seizures occurring at least 24 hours apart) [33] but that did not involve nodding of the head.

  • Colebunders, R., Nelson Siewe, F. J., & Hotterbeekx, A. (2018). Onchocerciasis-Associated Epilepsy, an Additional Reason for Strengthening Onchocerciasis Elimination Programs. Trends in Parasitology, 34(3), 208–216. https://doi.org/10.1016/j.pt.2017.11.009

2. Can the authors exclude neurocysticercosis in the nodding and OFE cohorts?

Response:

In this study, we did not assess for neurocysticercosis as a risk factor of NS and OFE. Indeed, neurocysticercosis is a significant contributor to epilepsy in low-income countries. However, previous brain imaging and postmortem studies did not show evidence for neurocysticercosis as a possible cause of nodding syndrome and onchocerciasis-associated epilepsy in northern Uganda  (Hotterbeekx et al., 2019; Winkler et al., 2013); Ogwang R et al.).

It cannot be excluded that in some persons with OFE, the epilepsy was caused by neurocysticercosis. However, in a previous case-control study in the study area, all nodding cases and all controls did not present Taenia solium antibodies (Foltz et al., 2013).

  •  Foltz, J. L., Makumbi, I., Sejvar, J. J., Malimbo, M., Ndyomugyenyi, R., Atai-Omoruto, A. D., Alexander, L. N., Abang, B., Melstrom, P., Kakooza, A. M., Olara, D., Downing, R. G., Nutman, T. B., Dowell, S. F., & Lwamafa, D. K. W. (2013). An Epidemiologic Investigation of Potential Risk Factors for Nodding Syndrome in Kitgum District, Uganda. PloS One, 8(6), e66419. https://doi.org/10.1371/journal.pone.0066419
  • Hotterbeekx, A., Lammens, M., Idro, R., Akun, P. R., Lukande, R., Akena, G., Nath, A., Taylor, J., Olwa, F., Kumar-Singh, S., & Colebunders, R. (2019). Neuroinflammation and Not Tauopathy Is a Predominant Pathological Signature of Nodding Syndrome. Journal of Neuropathology and Experimental Neurology, 78(11), 1049–1058. https://doi.org/10.1093/jnen/nlz090
  • Winkler, A. S., Friedrich, K., Velicheti, S., Dharsee, J., König, R., Nassri, A., Meindl, M., Kidunda, A., Müller, T. H., Jilek-Aall, L., Matuja, W., Gotwald, T., & Schmutzhard, E. (2013). MRI findings in people with epilepsy and nodding syndrome in an area endemic for onchocerciasis: An observational study. African Health Sciences, 13(2), 529. https://doi.org/10.4314/ahs.v13i2.51
  • Ogwang R, Ningwa A, Akun P, Bangirana P, Anguzu R, Mazumder R, Salamon N, Henning OJ, Newton CR, Abbo C, Mwaka AD, Marsh K, Idro R. Epilepsy in Onchocerca volvulus Sero-Positive Patients From Northern Uganda-Clinical, EEG and Brain Imaging Features. Front Neurol. 2021 Jun 3;12:687281.

3. Why did the authors not compare nodding, OFE and healthy controls together? This would be a more appropriate approach, especially since the control group came from the same study site. Moreover, multivariate regression analysis comparing nodding vs OFE, nodding vs healthy controls and OFE vs healthy control would be important to perform to analyse the differences between nodding and OFE. Finally, in my opinion it is redundant and not necessary to show the values of the healthy controls twice in the very same table (Table 2).

Response:

We compared nodding, OFE and health controls together. In multivariable regression analysis, we also compared nodding vs OFE, nodding vs healthy controls and OFE vs healthy control. And indeed, we found little difference between nodding vs OFE. The only analysis we have not shown, not to complicate the paper for the reader, is the comparison of nodding + OFE vs healthy controls.

We have now edited table 2 and show the values of health controls only once.

4. Can the authors explain why the age groups where separated in four different groups? What is the rational behind this separation? For example, why did the authors separate the age groups 14-15 and 16-17?

Response:

About 80% of the recruited study participants were aged between 14 to 17 years. Therefore, for analysing the data according to age groups, we separated this age group in two.

5. In regards to Ov16 antibody results, can the authors show the Ov16 IgG single to noise ratio and seropositive numbers in the different age groups? Is there an increase of Ov16 positivity with increasing age (due to increased duration of exposure to O. volvulus)?

Response:

Seropositivity also increased with increasing age: in the age group 6 – 13 years, 39 (60.9%) of the 64 were seropositive; in the age-group 14 – 15 years, 144 (76.2%) of the 189; in the age-group 16 -17 years, 163 (89.1%) of the 183; and in the age group 18 -24 years 23 (92.0%) of the 25.”

We now include this data in the paper.

 6. Besides the Ov16 results did the authors have data sets about Ov microfilaria counts, presence of nodules or data sets about clinical examination to elucidate active Ov infection. Presence of Ov16 antibodies do not predict if the patient was infected/have an ongoing infection since exposure to Ov without infection manifestation is also possible. If not this needs to be stated as limitation.

Response:

Thank you for this question. Indeed, we also assessed the participants for active onchocerciasis infection by using microscopy of skin-snip biopsies (from the iliac crests). We now include this data in the paper. There were only a few persons with a positive skin test; 13/154 (8.4%) among persons with OFE; 8/154 (5.2%) among persons with NS; and 2/ 153 (1.3%) among healthy controls. The low numbers of active onchocerciasis infections can be attributed to the rigorous onchocerciasis elimination programs (vector control and biannual mass distribution of ivermectin).

In the methods section (data collection), we now state: “From each participant, two skin-snips were obtained from the iliac crests to be examined for the presence of O. volvulus microfilariae.”

In the results section, we now state: “Onchocerca volvulus microfilariae were detected in skin snips of the minority of study participants: in 8.4% of persons with OFE, 5.2% of persons with NS and 1.3% in healthy controls (Table 2).”

The results of the skin snips have also been included in Table 2.

In the discussion section, we now state: “The low numbers of study participants with positive skin snips (presence of active onchocerciasis infections) can be attributed to the rigorous onchocerciasis elimination programs (vector control and biannual mass distribution of ivermectin) in the study area.

In the limitations, we state: “Finally, the assessment of onchocerciasis nodules among the study population was not consistent enough to be considered for analysis."

Reviewer 2 Report

This is a nested case control study of risk factors for nodding epilepsy and other forms of epilepsy compared to controls. The study is nested within a trials study which was set up in 2016/2017 and started afterwards. Results of that trial  study have not been released or published as far as known.

Given the nature of the study population and the settings, the set up of the current study is adequate. 

The results ar more or less in line with previous study, soms differences are apparent: no association with birth in IDP camps, rather before, and positive association with premature delivery. 

I have only a few remarks/questions that the authors may want to address: 

  1. Were persons recruited for the current nested study simultaneously with recruitment for the doxycycline trial, or sequentially?
  2. Were all persons recruited for and participating into the doxycycline trial als participating in the nested case control trial?
  3. In the discussion, the authors propose to start a trial of ivermectin in pregnant women with Onchocerca  volvulus infestation, despite the notion that this compound i contraindicated for use in pregnancy. Such a statement needs to be corroborated by a more detailed information about other background of the current guidelines and what is known about the teratogenic risks of ivermectin. 

Author Response

We thank the Reviewer for taking the time to review, comment, and give suggestions to this paper. Your comments have been valuable and enriching to the paper. Below are the responses to your comments and suggestions.

REVIEWER 2

This is a nested case control study of risk factors for nodding epilepsy and other forms of epilepsy compared to controls. The study is nested within a trials study which was set up in 2016/2017 and started afterwards. Results of that trial study have not been released or published as far as known.

Given the nature of the study population and the settings, the set up of the current study is adequate.

The results ar more or less in line with previous study, soms differences are apparent: no association with birth in IDP camps, rather before, and positive association with premature delivery.

I have only a few remarks/questions that the authors may want to address:

  1. Were persons recruited for the current nested study simultaneously with recruitment for the doxycycline trial, or sequentially?

Response:

The 154 Nodding syndromes (NS) participants in this case-control study were the first of the 230 NS cases recruited in the doxycycline trial.

Persons with other forms of epilepsy (OFE) were sequentially recruited after each case of NS.

The healthy controls were also sequentially recruited from the same communities as those persons with NS.

  1. Were all persons recruited for and participating into the doxycycline trial also participating in the nested case control trial?

Response:

Only the first 154 persons out of the 230 persons recruited for the doxycycline trial were selected to participate in the case-control study.

For more clarity, we state in the text that; “Between September 2016 and September 2017, the first 154 consecutive persons with NS, to be enrolled in the doxycycline clinical trial, were also recruited into a case-control study”.

  1. In the discussion, the authors propose to start a trial of ivermectin in pregnant women with Onchocerca volvulus infestation, despite the notion that this compound is contraindicated for use in pregnancy. Such a statement needs to be corroborated by a more detailed information about other background of the current guidelines and what is known about the teratogenic risks of ivermectin.

Response:

We agree that “to start a trial of ivermectin in pregnant women with Onchocerca volvulus infestation” needs more explanation. Therefore, we now omit this statement.

In the discussion, we now state, “If preterm birth related to an O. volvulus infection during pregnancy is a risk factor for NS and OFE, it may be advantageous to treat O. volvulus infected pregnant women with ivermectin. However, currently, pregnancy is a contraindication for ivermectin treatment. Therefore, additional studies are needed to investigate the safety and potentially beneficial effect of ivermectin in O. volvulus pregnant women.”

.

Round 2

Reviewer 1 Report

1) I thank the reviewer for answering the raised question. However, in regards to the several papers which were published within the last months by the group of Colebunders the definition of OFE and OAE remains unclear to me. The authors stated that "The term “Onchocerciasis Associated Epilepsy (OAE)” refers to the different forms of epilepsy (including Nodding syndrome and Nakalanga syndrome) that are epidemiologically linked to an Onchocerca volvulus infection and that meet specific criteria (Colebunders et al., 2018)." but that the specific criteria are not systemically assessed here. Can the authors specify what criteria they mean. To my knowledge Onchocerciasis-associated epilepsy (OAE) was defined as ≥2 seizures without any obvious cause, starting between the ages of 3–18 years in previously healthy persons who had resided for at least 3 years in the onchocerciasis endemic area should be available according to the method section and when patients were recruited from a clinical trial, since socio-demographic information and history of treatment/disease will be assessed during clinical trials. Moreover, skin snips and Ov16 were performed which lead to the conclusion that the study site is an onchocerciais endemic area. I am so critical about this fact since criteria for the different patient groups are in my opinion uncertain. I realized that also during the review process for some other papers from the group of Colebunders. In understand that information about health an clinical assessment of epilepsy/nodding is difficult to assess but for sophisticated analysis/comparison of groups clear definitions and inclusion/exclusion criteria need to be taken in consideration. Thus, I would like to ask the authors to define the different groups and clearly explain what criteria were not assessed also in comparison to the recently published papers within this special issue.

2) The missing diagnosis of neurocysticercosis needs to be included within the discussion section as limitation of the study.

4) In my opinion this explanation makes no sense from the biological point of view. Do the authors expect any differences between 14-15 and 16-17 years old that can affect the findings. What are the results when you split the group into 4 age groups or leave it as one group?

6) I thank the authors for the addition but in the discussion section the authors state: "We observed that O. volvulus infection was associated with developing both NS and OFE." (line 187-188)

Also in the conclusion section the authors state that O. volvulus infections associated with NS and OFE.

As mentioned before from the Ov16 results O. volvulus infection cannot be predicted and thus the authors need to state Ov seropositivity instead of infection .

Author Response

We thank you for your comments, and below are our responses to those comments and suggestions. 1) I thank the reviewer for answering the raised question. However, in regards to the several papers which were published within the last months by the group of Colebunders the definition of OFE and OAE remains unclear to me. The authors stated that "The term “Onchocerciasis Associated Epilepsy (OAE)” refers to the different forms of epilepsy (including Nodding syndrome and Nakalanga syndrome) that are epidemiologically linked to an Onchocerca volvulus infection and that meet specific criteria (Colebunders et al., 2018)." but that the specific criteria are not systemically assessed here. Can the authors specify what criteria they mean. To my knowledge Onchocerciasis-associated epilepsy (OAE) was defined as ≥2 seizures without any obvious cause, starting between the ages of 3–18 years in previously healthy persons who had resided for at least 3 years in the onchocerciasis endemic area should be available according to the method section and when patients were recruited from a clinical trial, since socio-demographic information and history of treatment/disease will be assessed during clinical trials. Moreover, skin snips and Ov16 were performed which lead to the conclusion that the study site is an onchocerciasis endemic area. I am so critical about this fact since criteria for the different patient groups are in my opinion uncertain. I realized that also during the review process for some other papers from the group of Colebunders. In understand that information about health an clinical assessment of epilepsy/nodding is difficult to assess but for sophisticated analysis/comparison of groups clear definitions and inclusion/exclusion criteria need to be taken in consideration. Thus, I would like to ask the authors to define the different groups and clearly explain what criteria were not assessed also in comparison to the recently published papers within this special issue. RESPONSE The protocol for this case-control study was developed, and data was collected before the OAE case definition was published. Therefore, we did not systematically ask how long the persons with epilepsy lived in the area, when they developed their first seizures, and whether their psychometric development was normal. Additionally, we did not ask questions to exclude other potential causes of epilepsy in onchocerciasis endemic areas ( meningitis, coma, measles, head trauma). The criteria for enrolment of OFE was that they only had to meet the ILAE criteria of epilepsy but without nodding of the head. This group of OFE, therefore, includes persons with epilepsy of different aetiology. However, the fact that this group presented similar risk factors to the group of nodding cases suggests that in the study area, many of the children in the age group of the nodding cases present as OAE. We now included in the discussion as a limitation: “Since this study was initiated before the definition of OAE was published, we did not systematically assess the study participants for all onchocerciasis-associated epilepsy (OAE) criteria. The assessment of onchocerciasis nodules among the study population was not consistently carried out to be considered for analysis.” 2) The missing diagnosis of neurocysticercosis needs to be included within the discussion section as limitation of the study. RESPONSE We now included in the discussion as limitations: “Finally, we cannot exclude that in some participants the epilepsy was caused by neuro-cysticercosis. However, previous brain imaging and post-mortem studies did not show evidence for neurocysticercosis as a possible cause of NS and OAE in northern Uganda [24,44,45]. Moreover, in a previous case-control study in the study area, all NS cases and all controls did not present Taenia solium antibodies [19].” 19. Foltz, J.L.; Makumbi, I.; Sejvar, J.J.; Malimbo, M.; Ndyomugyenyi, R.; Atai-Omoruto, A.D.; Alexander, L.N.; Abang, B.; Melstrom, P.; Kakooza, A.M.; et al. An Epidemiologic Investigation of Potential Risk Factors for Nodding Syndrome in Kitgum District, Uganda. PloS One 2013, 8, e66419, doi:10.1371/journal.pone.0066419. 24. Winkler, A.S.; Friedrich, K.; Velicheti, S.; Dharsee, J.; König, R.; Nassri, A.; Meindl, M.; Kidunda, A.; Müller, T.H.; Jilek-Aall, L.; et al. MRI Findings in People with Epilepsy and Nodding Syndrome in an Area Endemic for Onchocerciasis: An Observational Study. Afr. Health Sci. 2013, 13, 529, doi:10.4314/ahs.v13i2.51. 44. Hotterbeekx, A.; Lammens, M.; Idro, R.; Akun, P.R.; Lukande, R.; Akena, G.; Nath, A.; Taylor, J.; Olwa, F.; Kumar-Singh, S.; et al. Neuroinflammation and Not Tauopathy Is a Predominant Pathological Signature of Nodding Syndrome. J. Neuropathol. Exp. Neurol. 2019, 78, 1049–1058, doi:10.1093/jnen/nlz090. 45. Ogwang, R.; Ningwa, A.; Akun, P.; Bangirana, P.; Anguzu, R.; Mazumder, R.; Salamon, N.; Henning, O.J.; Newton, C.R.; Abbo, C.; et al. Epilepsy in Onchocerca Volvulus Sero-Positive Patients from Northern Uganda – Clinical, EEG and Brain Imaging Features. Front. Neurol. 2021, 12, doi:10.3389/fneur.2021.687281. 4) In my opinion this explanation makes no sense from the biological point of view. Do the authors expect any differences between 14-15 and 16-17 years old that can affect the findings. What are the results when you split the group into 4 age groups or leave it as one group? RESPONSE We agree, we now present the 4 age groups as stated and used in the protocol. We have re-run the logistic regression models using those age groups. There were slight but insignificant changes in the results, which we now present in table 1 and 3. It is these age groups that we based on while recruiting the 154 persons with NS. We then age-frequency matched the NS cases with 154 with OFE and 154 healthy community controls. 6) I thank the authors for the addition but in the discussion section the authors state: “We observed that O. volvulus infection was associated with developing both NS and OFE.” (line 187-188) RESPONSE We now state in the discussion that: “We observed that O. volvulus seropositivity was associated with both NS and OFE.” 6b.) Also, in the conclusion section the authors state that O. volvulus infections associated with NS and OFE. As mentioned before from the Ov16 results O. volvulus infection cannot be predicted and thus the authors need to state Ov seropositivity instead of infection RESPONSE The OV16 Elisa test is very specific, and OV16 seropositivity means there has been an O. volvulus infection in the past. We state in the conclusion that “Our findings support the relationship between O. volvulus seropositivity and the development of NS and OFE in the northern Uganda study area.”
